# Association of Thermoresponsive Diblock Copolymer PDEGMA-*b*-PDIPAEMA in Aqueous Solutions: The Influence of Terminal Groups

**DOI:** 10.3390/polym16152102

**Published:** 2024-07-24

**Authors:** Adam Škorňa, Dimitrios Selianitis, Stergios Pispas, Miroslav Štěpánek

**Affiliations:** 1Department of Physical and Macromolecular Chemistry, Faculty of Science, Charles University, Hlavova 2030, 128 40 Prague, Czech Republic; skorna.adam@gmail.com; 2Theoretical & Physical Chemistry Institute, National Hellenic Research Foundation, 48 Vassileos Constantinou Avenue, 11635 Athens, Greece; dimitrissel404@gmail.com (D.S.); pispas@eie.gr (S.P.)

**Keywords:** thermoresponsive polymers, pH-responsive polymers, block copolymers, association

## Abstract

Aqueous solutions of a thermoresponsive diblock copolymer poly(di-[ethylene glycol] methyl ether methacrylate)-*b*-poly(2-[diisopropylamino] ethyl methacrylate) (PDEGMA-*b*-PDIPAEMA) were studied by static, dynamic and electrophoretic light scattering, small-angle X-ray scattering and differential scanning calorimetry. Thermoresponsive behavior of PDEGMA-*b*-PDIPAEMA was investigated at two pH values, pH = 2, at which the terminal carboxylic group of the PDEGMA chain and the PDIPAEMA block are protonated, and pH = 7, where the carboxyl terminal group is ionized while the PDIPAEMA block is partially deprotonated and more hydrophobic. Both at pH = 2 and 7, PDEGMA-*b*-PDIPAEMA copolymer underwent extensive association (the size of the aggregates was between 100 and 300 nm), indicating strong interchain interactions. While the measurements confirmed thermoresponsive behavior of PDEGMA-*b*-PDIPAEMA at pH = 7, no changes in the association with temperature were observed at pH 2 as the thermoresponsivity of PDEGMA was suppressed by hydrogen bonding between carboxylic groups and PDEGMA segments, as well as due to the increased hydrophilicity of the PDIPAEMA block. Fluorescence measurements with pyrene as a fluorescent probe showed that both at pH = 2 and pH = 7 the associates were able to solubilize hydrophobic substances.

## 1. Introduction

The ability of block copolymers (BCs) to form self-assembled core/shell nanoparticles [1,2] such as micelles or vesicles (polymersomes) in selective solvents has been exploited in many applications in medicine (controlled drug delivery [3,4,5,6], theranostic applications [7,8]) or in technology (nanoreactors [9,10,11], emulsifiers in Pickering emulsions [12]).

For drug delivery applications, it is often advantageous if BCs forming the nanoparticles in aqueous media respond to external stimuli such as temperature [13,14,15], pH [16,17,18], ionic strength [18] or the presence of specific molecules [19]. A change in these stimuli can either render core-forming blocks of the nanoparticle soluble [16], thus triggering disruption of nanoparticles and release of the cargo or disrupt the interactions between the cargo and the core-forming blocks [17,19], thus enabling its release by diffusion. Combining responsive blocks allows for the design of multiple stimuli-responsive block copolymers with complex phase behavior in solutions.

Specifically, block copolymers which are able to swap soluble and insoluble blocks upon change of conditions and form reverse core/shell nanoparticles have been referred to as “schizophrenic” [20,21,22]. These swaps occur spontaneously upon change of external stimuli, which distinguishes schizophrenic block copolymers from many amphiphilic block copolymers with kinetically frozen hydrophobic cores in aqueous solutions. Transitions of the blocks from hydrophilic to hydrophobic and vice versa may lead to extensive morphological transitions [21]. Typically, schizophrenic block copolymers consist of at least one thermoresponsive block [22] often in combination with weak polyelectrolyte blocks which allow for triggering micellization of the copolymer by changes in pH.

In this work, we report on the association of a schizophrenic diblock copolymer, poly(di-[ethylene glycol] methyl ether methacrylate)-*b*-poly(2-[diisopropylamino] ethyl methacrylate), PDEGMA-*b*-PDIPAEMA (Figure 1), in aqueous solutions. While PDEGMA [23] is a thermoresponsive polymer with the lower critical solution temperature (LCST) below 303 K, the PDIPAEMA [24] behavior in aqueous solution is more complex as its hydrophobicity can be tuned by both pH (the p*K* of the diisopropylamino group is 6.5) and temperature (the cloud point is pH dependent).

In the previous papers, the association behavior of PDEGMA-*b*-PDIPAEMA was studied in aqueous solution [25] and in air/water interfaces [26]. In this work, we study the solution behavior of PDEGMA-PNIPAM in more detail, using static and dynamic light scattering, SAXS, DSC and fluorometry with pyrene as a hydrophobic fluorescent probe. In our earlier studies, we showed that terminal groups from the chain transfer agent (CTA) used for RAFT polymerization of block copolymers had influence on their self-assembly behavior [27]. Therefore we also focused on this influence in the case of PDEGMA-*b*-PDIPAEMA, synthesized using (2-cyano-2-[(dodecylsulfanylthiocarbonyl) sulfanyl] propane) as CTA. For that reason, we study the thermoresponsive association behavior at pH = 2, when PDIPAEMA block is fully protonated and so is the terminal carboxylic group, and at pH = 7, when the degree of PDIPAEMA protonation is only ca. 50% and the terminal carboxylic group is ionized.

## 2. Materials and Methods

### 2.1. Materials

Aqueous solutions were prepared using deionized water obtained from MicroPure Water Purification System (Thermo Scientific, Langenselbold, Germany). Hydrochloric acid (analytical grade, 35% *w/w*) was purchased from LachNer, Czech Republic, methanol (≥99.8%, HPLC grade) and pyrene (≥99.0%, for fluorescence grade) were purchased from Sigma-Aldrich (St. Louis, MI, USA). IUPAC buffers pH 4.005 and 7.000 (Radiometer Analytical, Villeurbane, France) were used for calibration of a pH electrode. Reagents used for PDEGMA-*b*-PDIPAEMA synthesis are listed in [25].

### 2.2. Light Scattering (LS)

The static and dynamic light LS measurements were conducted simultaneously, collecting both time-averaged static scattering intensities, *I*(*q*) = 〈*I*(*q*,*t*)〉 and intensity autocorrelation functions, *g*^(2)^(*q*,*τ*) = 〈*I*(*q*,*t*)*I*(*q*,*t* + *τ*)〉/〈*I*(*q*,*t*)〉^2^, where *τ* is the lag time and *q* is the scattering vector magnitude. The used setup consisted of a ALV CGS-3 light scattering photometer (ALV, Langen, Germany), a 100 mW diode-pumped solid-state laser with wavelength *λ* = 660 nm, two high quantum efficiency avalanche photodiode photon counting modules operated in a pseudo cross-correlation mode and an ALV 5004 multiple-tau digital correlator. The *g*^(2)^(*q*,*τ*) functions were fitted using an inverse Laplace transform using a constrained regularization algorithm (CONTIN).

The measurements were conducted in cylindrical glass cells in the angular range of 30°–150° (corresponding to the *q* range 6.6 to 24.5 μm^−1^ in aqueous solutions, where *q* is the magnitude of the scattering vector, *q* = (4π*n*/*λ*) sin(*θ*/2), *q* being scattering angle and *n* solvent refractive index) with the angular step of 10° and temperature varying from 15 to 35 °C with 2 °C steps in sealed test tubes with 10 mm diameter. The calibration to the absolute scattered intensity scale was performed using toluene as a standard, assuming the Rayleigh ratio of 1.14 × 10^−5^ cm^−1^. An external bath circulator connected to the sample cell housing system was used to control the temperature.

### 2.3. Electrophoretic Light Scattering

Temperature dependences of zeta potentials were measured on a Zeta Sizer Nano ZS (Malvern, UK) equipped with a Peltier module for temperature control, using dip cells with palladium electrodes. Zeta potentials ζ were calculated from electrophoretic mobilities *μ* using the Henry equation in the Smoluchowski limit, *ζ* = *μη*/*ε*, where *ε* is solvent permittivity. The values are averages of 5 measurements, each consisting of 3–10 runs.

### 2.4. Small-Angle X-ray Scattering

SAXS experiments were performed at the BIOCEV centre (Vestec, Czech Republic) with a SAXSpoint 2.0 instrument (Anton Paar, Graz, Austria) equipped with an Excillum MetalJet C2 X-ray source and with an Eiger R 1M detector (Dectris, Baden-Daettwil, Switzerland). The active area of the detector and the pixel size were 79.9 mm × 77.2 mm and 75 μm × 75 μm, respectively, the X-ray source wavelength was λ = 0.1348 nm. Measurements were conducted at 20 °C in a 1 mm quartz capillary filled using an automatic sampler, at a sample-to-detector distance of 0.82 m, corresponding to the *q* vector range from 0.04 to 4 nm^−1^. SAXS curves were obtained by azimuthal averaging of scattering patterns for the sample and the solvent (either water or 0.01 M HCl) and subtracting the solvent intensities from those for the sample.

### 2.5. Differential Scanning Calorimetry (DSC)

DSC experiments were performed using a Nano DSC differential scanning calorimeter (TA Instruments, New Castle, DE, USA), equipped with platinum reference and sample cells with a volume of 650 μL. The scanning was carried out under the pressure of 3 atm in the temperature range from 288 to 333 K at a rate of 1 K/min. Data were analyzed using the NanoAnalyze 3.8.0 software.

### 2.6. Fluorometry

Fluorescence emission spectra were measured at 25 °C using a Fluorolog FL3-22 fluorometer (Horiba, Vénissieux, France) equipped with a 450 W high-pressure xenon lamp in 1 cm rectangular quartz cuvettes at the excitation wavelength 315 nm. The excitation and emission bandpasses were set to 5 nm and 2 nm, respectively.

### 2.7. PDEGMA-b-PDIPAEMA Synthesis and Characterization

In a previous work [25], some of us reported the synthesis of several PDEGMA-*b*-PDIPAEMA copolymers with different chemical composition of the blocks and molecular weights utilizing the reversible addition fragmentation-chain transfer (RAFT) polymerization technique. Following similar synthesis procedures, a poly (diethylene glycol methacrylate) (PDEGMA) homopolymer was synthesized in 1,4-dioxane using 2,2’-azobis(isobutyronitrile) (AIBN) as the radical initiator and 4-cyano-4-[(dodecylsulfanylthiocarbonyl) sulfanyl)] pentanoic acid (CDTP) as the chain transfer agent. The PDEGMA homopolymer was isolated by precipitation in hexane (twice) and after drying was utilized as the macro-CTA agent for the RAFT polymerization of diisopropyl aminoethyl methacrylate (DIPAEMA), in the presence of additional AIBN as the initiator and 1,4-dioxane as the solvent. The polymerizations for the synthesis of PDEGMA and PDEGMA-*b*-PDIPAEMA were carried out at 70 °C for 6 and 5 h, respectively. Molecular characterization (the details can be found in ref. [25]) of the diblock copolymer was accomplished by means of size exclusion chromatography (SEC) in THF, containing 5% *v/v* triethylamine using narrow polystyrene standards, ^1^H-NMR spectroscopy (in CDCl_3_) and ATR-FTIR spectroscopy. The results of the analysis are presented in Figure 1. The weight-averaged molecular weight, dispersity index and the weight fraction of the PDEGMA block were 10.9 kg mol^−1^, 1.38 and 32%, respectively.

### 2.8. Preparation of PDEGMA-b-PDIPAEMA Solutions

The copolymer solutions at were prepared by direct dissolution of PDEGMA-*b*-PDIPAEMA copolymer in deionized water (pH = 7) or in 0.01 M aqueous HCl (pH = 2) at room temperature and stored at 5 °C. pH of the solutions was measured with an Orion Star A211 pH meter (Thermo Scientific, Beverly, Massachusetts, USA), equipped with a Radiometer pHC3006-9 (Radiometer Analytical, Villeurbane, France) combined glass electrode. The copolymer concentrations for DSC, SAXS and LS measurements were 15, 10 and 1 mg/mL, respectively. The solutions for fluorometry were prepared by mixing 5 μL of 0.5 mmol/L solution of pyrene in methanol to 5 mL of 1 mg/mL aqueous copolymer solution. The solutions were then left to stand overnight for equilibration.

## 3. Results and Discussion

DSC measurements (Figure 2) in the temperature range 280–330 K showed that the thermoresponsive behavior of PDEGMA-*b*-PDIPAEMA copolymer in aqueous solution was pH dependent. At pH = 7, the sample exhibited two exothermic transitions with onsets at 293 K and 319 K. While the former was caused by the dehydration of the PDEGMA block (that is, the loss of water molecules solvating the PDEGMA chains) above its LCST, the latter was most likely caused by the respective transition of the PDIPAEMA block, the cloud point of which was reported to be 312 K at pH = 6 [26]. In the further study we focused only on the transition of PDEGMA at 293 K.

On the other hand, no phase transitions were observed by DSC at pH = 2. In the case of PDIPAEMA block, this is most likely caused by low pH which shifted the transition temperature out of the investigated temperature range. The ability of PDIPAEMA to undergo conformational changes is also dependent on intrachain electrostatic repulsion—fully protonated PDIPAEMA at pH = 2 is more rigid as compared with PDIPAEMA at pH = 7 with degree of protonation about 0.5. In the case of PDEGMA, the interpretation of the lack of thermoresponsivity at low pH is more difficult. Before we get to it, let us discuss the results of light scattering (LS) measurements.

Static LS measurements (Appendix A) in the temperature range 288–307 K corroborated the calorimetric data. The forward scattering intensities, *I*(0), and the radii of gyration, *R*_g_, obtained by fitting the scattering curves *I*(*q*) in the low *q* range by the Guinier equation
(1)$$\ln\frac{I(q)}{I(0)}=-\frac{1}{3}R_{g}^{2}q^{2}$$
are plotted in Figure 3 and Figure 4. At pH = 7, both *I*(0) and *R*_g_ steeply increased around 293 K, indicating association of PDEGMA-*b*-PDIPAEMA chains above the LCST of PDEGMA. On the other hand, at pH = 2 both *I*(0) and *R*_g_ remained almost constant in the studied temperature range, at about 0.1 dm^−1^ and 175 nm, respectively, indicating that the copolymer formed associates even below the LCST of PDEGMA and those associates did not change significantly with passing through the LCST transition of PDEGMA. This behavior was most likely caused by the formation of complexes stabilized by hydrogen bonds between COOH and CH_2_CH_2_O groups, similarly to complexes formed by poly(ethylene oxide) with poly(acrylic) [28] or poly(methacrylic acid) [29] in aqueous solutions at low pH. Such interaction would render PDEGMA domains more compact and/or hindered, which would result in limited ability of PDEGMA to undergo conformational rearrangement upon dehydration.

At both pH = 2 and pH = 7, the size of the aggregates was much larger than what would correspond to core/shell micelles, considering the contour length of the diblock chains. We would like to stress that this behavior was not caused by the presence of kinetically trapped aggregates in the solid polymer. The same results were obtained after dissolution of the sample in tetrahydrofuran (THF), mixing with water and subsequent removal of THF by dialysis. This means that the observed association of PDEGMA-*b*-PDIPAEMA upon dissolution in water occurred spontaneously and either it dominated over formation of micelles with segregated PDEGMA and PDIPAEMA blocks or PDEGMA-*b*-PDIPAEMA micelles or micelle-like aggregates formed first and then they still underwent secondary association to larger particles. As both blocks are short (18 and 35 monomeric units, respectively), they were probably not strongly segregated and the formed domains did not have well-defined interfaces. It is likely that PDEGMA-*b*-PDIPAEMA with longer blocks would be more apt to micellization rather than to formation of large associates, however, the confirmation would require a systematic study with a series of PDEGMA-*b*-PDIPAEMA samples varying in molecular weight in a rather wide range.

In general, the formation of large associates in aqueous solutions of water-soluble polymers is not surprising. It was reported several times that long range electrostatic interactions between polyelectrolyte chains and their shared counterion clouds in low ionic strength aqueous solutions caused formation of concentrated polyelectrolyte domains, diffusion of which appeared in DLS as a slow mode, in addition to a fast mode corresponding to the diffusion of individual polyelectrolyte chains [30]. Similarly, neutral hydrophilic polymers such as poly(ethylene oxide) were also observed to form aggregates in aqueous solutions, stabilized by a network of hydrogen bonds [31]. However, in the case of PDEGMA-*b*-PDIPAEMA as a diblock copolymer with responsivity to both temperature and pH, it is difficult to predict how these responsivities affect the behavior of the associates. Moreover, it is also necessary to keep in mind that terminal dodecyl groups of PDIPAEMA may further contribute to stabilization of PDEGMA-*b*-PDIPAEMA associates by hydrophobic effect [27].

Further information about the structure of the associates can be gained from the comparison of their *R*_g_ with their hydrodynamic sizes, extracted from dynamic LS measurements. However, to do so, it is necessary to use hydrodynamic radii calculated from real diffusion coefficients. Apparent diffusion coefficients for large non-uniform scatterers are generally *q*-dependent, reflecting differences in dynamics of fluctuations in the solution at the given length scale and their value have to be extrapolated to zero *q* to provide “true” diffusion coefficient corresponding to translational diffusion of the scatterers. As the CONTIN fits of the autocorrelation functions revealed monomodal distributions of apparent hydrodynamic radii (Appendix A), normalized DLS autocorrelation functions $g^{\left(1\right)}(q,\tau )$, related to the obtained $g^{\left(2\right)}(q,\tau )$ functions by the Siegert relation, $g^{\left(2\right)}\left(q,\tau \right)=1+\beta {\left|g^{\left(1\right)}(q,\tau )\right|}^{2}$, where *β* is the coherence factor, could be processed simply using the 2nd order cumulant expansion:(2)$$\ln g^{\left(1\right)}(q,\tau )=-\Gamma_{1}\left(q\right)\tau +\frac{1}{2}\Gamma_{2}\left(q\right)\tau^{2}$$
where, Γ_1_(*q*), Γ_2_(*q*) are the 1st and 2nd cumulants (1st and 2nd moments of the distribution function of relaxation rates). The hydrodynamic radii, *R*_H_, were then calculated using the Stokes-Einstein formula, *R*_H_ = *kT*/6*πηD*, where *k* is the Boltzmann constant, *η* is solvent viscosity and *D* is diffusion coefficient, obtained by linear extrapolation of apparent diffusion coefficients, *D*_app_(*q*) = Γ_1_(*q*)/*q*^2^ , to zero *q*:(3)$$\frac{\Gamma_{1}\left(q\right)}{q^{2}}=D\left(1+CR_{g}^{2}q^{2}\right)$$
where *C* is a constant generally dependent on size, shape dispersity, and internal dynamics of the scatterers. Such values represented true *z*-averaged diffusion coefficient and as such they could be used for calculation of reliable *R*_g_/*R*_H_ values.

While associates formed at pH = 2 were compact as indicated by the estimated values of *R*_g_/*R*_H_ ratios, being lower than 0.78, the value for uniform hard spheres (Figure 5), those formed at pH = 7 were much looser with *R*_g_/*R*_H_ > 1. This observation can be explained by the formation of hydrogen bonds between terminal COOH groups and CH_2_CH_2_O groups of PDEGMA which promoted formation of compact PDEGMA domains in the associates. Unlike pH = 2, at pH = 7 the ratio was also slightly decreasing with increasing temperature as a result of decreasing hydration of copolymer chains.

To investigate the differences in electrostatic interactions in PDEGMA-*b*-PDIPAEMA associates between pH = 2 and pH = 7, we conducted electrophoretic light scattering measurements. The results are shown in Figure 6. Surprisingly, at pH = 7 zeta potential is negative despite strongly positive net charge at this pH (one COO^−^ group per ca. 17 NH_3_^+^ groups considering the degree of PDIPAEMA protonation being approximately 0.5). However, it is necessary to keep in mind that *ζ* values are governed by surface charges. At pH = 7, the lower degree of protonation of PDIPAEMA supports hydrophobic association of PDIPAEMA blocks promoted by terminal dodecyl groups and segregated PDEGMA blocks with terminal COO^−^ groups then surround these domains. We would like to stress that electrostatic interaction between partially protonated PDIPAEMA and COO^−^ groups is of the same strength as that of PDIPAEMA with its small monovalent counterions (OH^−^) and since the number of counterions exceeds that of COO^−^ by an order of magnitude, formation of an electrostatic complex between PDIPAEMA and PDEGMA is unlikely; formation of electrostatic complexes require two oppositely charged polyelectrolytes or polyelectrolyte and oppositely charged multivalent ions as the complexation is accompanied by entropic gain from released small counterions upon formation of the complex [32].

Further insight on the internal structure of the aggregates was obtained from SAXS measurements (Figure 7), covering length scales in the range from nm units to tens of nm. In the *q* range 0.07–4 nm^−1^, the SAXS curves were fitted by the scattering function
(4)$$I\left(q\right)=I_{0}q^{\alpha }+\frac{I_{1}}{1+\xi^{2}q^{2}}$$
where *I*_0_ and *I*_1_ are the pre-factors, *α* is the power law exponent and *ξ* is the correlation length. The power-law term dominated the scattering behavior for *q* < 0.4 nm, corresponding to the length scales 15–90 nm. The exponent *α* < −3 suggested that the associates were homogeneous at those length scales and scattering occurred at interfaces. The associates formed at pH = 2 showed the Porod scattering with *α* = −4, indicating a flat interface, while at pH = 7 the exponent was higher, *α* = −3.2, thus indicating a surface fractal character of the interface.

For *q* > 0.4 nm, the differences in scattering of the associates between pH = 2 and pH = 7 were more pronounced. While at pH = 7 the scattering already was at the background level, at pH = 2 it exhibited significant fluctuations, which could be fitted with the Ornstein-Zernike function with the correlation length *ξ* = 0.82 nm. This behavior could be caused by inhomogeneities in the associates at the length scale of units of nm (compact PDEGMA domains interconnected by hydrogen bonds vs. loose domains formed by stretched fully protonated PDIPAEMA chains) or by the scattering from single PDEGMA-*b*-PDIPAEMA chains or smaller associates which coexist with the large associates in the solution.

To check if the associates were able to solubilize hydrophobic substances, we mixed PDEGMA-*b*-PDIPAEMA solution with saturated aqueous solution of pyrene and after equilibration we measured the pyrene emission spectrum which is sensitive to the microenvironment polarity (Figure 8). Both at pH = 2 and pH = 7, the pyrene fluorescence was quenched but also the probe experienced less polar microenvironment as indicated by the decrease in the ratio of the first (372 nm) to the third (383 nm) vibronic bands, *I*_1_/*I*_3_ (about 1.3 in copolymer solutions vs. 1.7 in water). This indicated that pyrene was solubilized in the associates both at pH = 2 and pH = 7. On the other hand, the difference in the degree of protonation of the PDIPAEMA block (that is 1.0 at pH = 2, vs. 0.5 at pH = 7) had no significant influence on local polarity of pyrene microenvironment in the associates. It is necessary to keep in mind that the obtained *I*_1_/*I*_3_ ratio is an average over various domains in the associates (with non-negligible contribution from free pyrene in the bulk solution) and that major contribution to hydrophobicity of the associates originated most likely from terminal hydrophobic groups, independently of protonation of PDIPAEMA.

## 4. Conclusions

We have examined the thermoresponsive behavior of PDEGMA-b-PDIPAEMA copolymer in aqueous solutions at pH = 2, with protonated terminal carboxylic groups at the PDEGMA block, and at pH = 7, with ionized carboxylic groups. We found that while at pH 7 the copolymer exhibited thermoresponsivity and underwent pronounced association which manifested itself in the increase of scattering intensity and gyration and hydrodynamic radii of the associates at temperatures larger than 293 K, no response to temperature changes was observed at pH = 2, most likely because of the formation of inter- and intrachain hydrogen bonds between terminal COOH groups and CH_2_CH_2_O units of the PDEGMA chains (Figure 2) which decreased the chain mobility and thus hindered their response to dehydration. Additionally, PDIPAEMA blocks were fully protonated at this pH which should further suppress association induced by the collapse of PDEGMA blocks.

Fluorescence measurements with pyrene as a hydrophobic polarity-responsive probe showed that PDEGMA-*b*-PDIPAEMA associates formed spontaneously upon the copolymer dissolution in water were able to solubilize hydrophobic substances in aqueous environment. The latter finding is important in the context of drug delivery potential applications because the preparation of the associates does not require using organic co-solvents as is the case of polymeric nanoparticles prepared by nanoprecipitation [33].

## Data Availability

Data available upon request.

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
