# Peer review of "Association of Thermoresponsive Diblock Copolymer PDEGMA-b-PDIPAEMA in Aqueous Solutions: The Influence of Terminal Groups"

_polymers, 2024, doi:10.3390/polym16152102_

Round 1

Reviewer 1 Report

Comments and Suggestions for Authors

In the submitted study, the authors investigated the solution behavior of PDEGMA-PNIPAM in more detail, using static and dynamic light scattering, SAXS, DSC, and fluorometry with pyrene as a hydrophobic fluorescent probe. Additionally, to investigate the terminal groups from the chain transfer agent (CTA) influence on their self-assembly behavior, they synthesize (2-cyano-2-[(dodecylsulfanylthiocarbonyl) sulfanyl] propane) as CTA. Finally, the thermoresponsive association behavior at pH 2 and 7 was also discussed, so the work looks interesting and can be published after providing the major issues:

1. In the abstract, the first statement is too long and hard to understand for the reader.

2. Keywords are also very long; for example, thermoresponsive block copolymer should be written separately.

3. In scheme 1, showing the structure of PDEGMA-PDIPAEMA, how does the author confirm the structure? If it's reported, then provide references or HNMR. 

4. The formation of hydrogen bonds between COOH most likely caused this behavior and CH2CH2O groups, which led to the formation of more compact PDEGMA domains with limited ability of conformational rearrangement upon dehydration, but how does the author confirm this formation of hydrogen bonds between COOH. Reference is needed.

5. At pH 2, when PDIPAEMA is fully protonated, it may undergo hydrophobic association because of the terminal dodecyl group, explained the PDIPAEMA protonation according to the reported literature for a better understanding.  

6. The PDEGMA-PDIPAEMA formed micelles with segregated PDEGMA and PDIPAEMA blocks, but why do they still undergo secondary association to larger particles? Please explain.

Comments on the Quality of English Language

Acceptable

Author Response

In the submitted study, the authors investigated the solution behavior of PDEGMA-PNIPAM in more detail, using static and dynamic light scattering, SAXS, DSC, and fluorometry with pyrene as a hydrophobic fluorescent probe. Additionally, to investigate the terminal groups from the chain transfer agent (CTA) influence on their self-assembly behavior, they synthesize (2-cyano-2-[(dodecylsulfanylthiocarbonyl) sulfanyl] propane) as CTA. Finally, the thermoresponsive association behavior at pH 2 and 7 was also discussed, so the work looks interesting and can be published after providing the major issues:

  1. In the abstract, the first statement is too long and hard to understand for the reader.

As suggested, we modified the abstract and divided the first long sentence in two .

  1. Keywords are also very long; for example, thermoresponsive block copolymer should be written separately.

We modified the keywords accordingly.

  1. In scheme 1, showing the structure of PDEGMA-PDIPAEMA, how does the author confirm the structure? If it's reported, then provide references or HNMR. 

In the revised manuscript, we provide the 1H NMR spectrum with peak assignments.

  1. The formation of hydrogen bonds between COOH most likely caused this behavior and CH2CH2O groups, which led to the formation of more compact PDEGMA domains with limited ability of conformational rearrangement upon dehydration, but how does the author confirm this formation of hydrogen bonds between COOH. Reference is needed.

As suggested, we provide two references of papers dealing with complexes of poly(ethylene oxide) and polyacrylic or polymethacrylic acid stabilized by hydrogen bonds.

  1. At pH 2, when PDIPAEMA is fully protonated, it may undergo hydrophobic association because of the terminal dodecyl group, explained the PDIPAEMA protonation according to the reported literature for a better understanding.

In the revised manuscript, we discuss the influence of dodecyl terminal groups more thoroughly; a reference to our earlier study dealing with this influence (a double hydrophilic diblock copolymer with terminal dodecyl group from RAFT) is provided. 

  1. The PDEGMA-PDIPAEMA formed micelles with segregated PDEGMA and PDIPAEMA blocks, but why do they still undergo secondary association to larger particles? Please explain.

In the revised manuscript, we discuss this problem more in detail. Even after segregation of PDEGMA and PDIPAEMA blocks into domains which would eventually lead to the formation of micelles, the domains still exhibit attraction mediated either electrostatically by shared cloud of counterions in the case of PDIPAEMA, which was observed for various polyelectrolytes (the slow mode in DLS) or, in the case of PDEGMA, by a network of hydrogen bonds in water, which was observed for poly(ethylene oxide). The fluctuations on the scale of units of nm observed by SAXS may be attributed to core-shell structure preserved in the large aggregates, although it is necessary to admit that these fluctuations may originate from free chains or small clusters in the bulk solution coexisting with large associates. Discerning between these two cases would require investigating dynamics of these short range fluctuations for example by neutron spin echo spectroscopy.  

Reviewer 2 Report

Comments and Suggestions for Authors

The manuscript presents a study on the association behavior of the thermoresponsive diblock copolymer PDEGMA-PDIPAEMA in aqueous solutions, emphasizing the influence of terminal groups.

The innovative aspect of this study lies in its focus on how the terminal carboxyl groups from the RAFT chain transfer agent impact the self-assembly behavior of the copolymers. Additionally, the examination of PDEGMA-PDIPAEMA under varying pH and temperature conditions to understand their aggregation behavior and responsiveness offers valuable insights for designing multi-stimuli responsive polymers.

However, there are several key points that need to be addressed before it can be considered for publication.

  1. Fundamental Reason for Lack of Thermal Response at pH 2: The manuscript attributes the absence of thermal responsiveness at pH 2 to hydrogen bonding between the terminal carboxyl groups and CH2CH2O units. While this is a valid point, it is crucial to highlight that at pH 2, the PDIPAEMA block is possibly protonated, which increases its hydrophilicity. This increased hydrophilicity likely prevents the copolymer from forming well-defined aggregates, fundamentally contributing to the lack of temperature responsiveness. This explanation should be clearly stated and discussed in the manuscript.

  2. Experimental Detail Enhancement and Comprehensive Data Analysis: The manuscript would benefit from more detailed descriptions of the experimental setups and conditions, particularly for the light scattering and SAXS measurements. Providing specifics such as scattering angles, temperature control methods, and calibration procedures would enhance the reproducibility and understanding of the experiments. Additionally, the data analysis sections should include a more in-depth discussion of how various parameters (e.g., radius of gyration, hydrodynamic radius) were derived and interpreted, especially in the context of different pH conditions. The authors should also enhance the data analysis for Figures 2, 3, and 4 to provide a clearer understanding of the copolymer’s behavior.

  3. Fluorescence Data Explanation: Figure 6 presents fluorescence data that does not show significant differences between the different pH values. The authors should provide an explanation for this observation, as it contrasts with the expected influence of pH on the copolymer's assembly behavior and microenvironment characteristics.

  4. Molecular Weight Influence: It would be valuable to consider whether the molecular weight of the block copolymer affects its aggregation behavior and thermal responsiveness. The manuscript should discuss this aspect or suggest further studies to explore it.

  5. Different End Groups and Functional Groups: Currently, the polymers have carboxyl end groups. It would be interesting to explore how amine end groups, or other functional groups with different pKa values and polarities, might influence the polymer's behavior. The authors should consider this aspect or suggest a systematic study to investigate the effects of different terminal groups.

Author Response

The manuscript presents a study on the association behavior of the thermoresponsive diblock copolymer PDEGMA-PDIPAEMA in aqueous solutions, emphasizing the influence of terminal groups. The innovative aspect of this study lies in its focus on how the terminal carboxyl groups from the RAFT chain transfer agent impact the self-assembly behavior of the copolymers. Additionally, the examination of PDEGMA-PDIPAEMA under varying pH and temperature conditions to understand their aggregation behavior and responsiveness offers valuable insights for designing multi-stimuli responsive polymers. However, there are several key points that need to be addressed before it can be considered for publication.

Fundamental Reason for Lack of Thermal Response at pH 2: The manuscript attributes the absence of thermal responsiveness at pH 2 to hydrogen bonding between the terminal carboxyl groups and CH2CH2O units. While this is a valid point, it is crucial to highlight that at pH 2, the PDIPAEMA block is possibly protonated, which increases its hydrophilicity. This increased hydrophilicity likely prevents the copolymer from forming well-defined aggregates, fundamentally contributing to the lack of temperature responsiveness. This explanation should be clearly stated and discussed in the manuscript.

We thank the reviewer for pointing out this issue. In the revised manuscript, we pay much more attention to the influence of the degree of PDIPAEMA protonation on the association behavior. The problem with explaining the lack of thermal responsivity by increased electrostatic repulsion between PDIPAEMA blocks is such that it would explain the lack of responsivity followed by light scattering (suppressed aggregation) but less so in the case of DSC because if conformational rearrangement of PDEGMA chain occurred, it should cause an observable heat effect regardless of whether it is followed by association or not. On the other hand, the lack of thermal response to higher temperature transition (the onset at 319 K) is clearly caused by increased protonation of PDIPAEMA which shifts its LCST. 

Experimental Detail Enhancement and Comprehensive Data Analysis: The manuscript would benefit from more detailed descriptions of the experimental setups and conditions, particularly for the light scattering and SAXS measurements. Providing specifics such as scattering angles, temperature control methods, and calibration procedures would enhance the reproducibility and understanding of the experiments. Additionally, the data analysis sections should include a more in-depth discussion of how various parameters (e.g., radius of gyration, hydrodynamic radius) were derived and interpreted, especially in the context of different pH conditions. The authors should also enhance the data analysis for Figures 2, 3, and 4 to provide a clearer understanding of the copolymer’s behavior.

In the revised manuscript, we expanded both the experimental section describing scattering measurements and data processing and the parts of results and discussion concerning the analysis of light scattering data. We hope that this addition will make the text clearer for the reader. 

Fluorescence Data Explanation: Figure 6 presents fluorescence data that does not show significant differences between the different pH values. The authors should provide an explanation for this observation, as it contrasts with the expected influence of pH on the copolymer's assembly behavior and microenvironment characteristics.

We added an explanation of that observation to the revised manuscript. In brief, the most likely reason why I1/I3 ratio does not differ significantly between pH 2 and pH 7 is that the dominant contribution to pyrene response to nonpolar microenvironment originates from its interaction with terminal dodecyl groups and this might be essentially the same both for pH 2 and pH 7.

Molecular Weight Influence: It would be valuable to consider whether the molecular weight of the block copolymer affects its aggregation behavior and thermal responsiveness. The manuscript should discuss this aspect or suggest further studies to explore it.

We thank the reviewer for this suggestion. In the revised manuscript, we suggest such a study and mention that we suppose that low molecular weight of the used PDEGMA-b-PDIPAEMA sample can be a reason why the copolymer forms the observed large associates consisting of multiple PDEGMA and PDIPAEMA domains instead of regular core-shell micelles with well-defined interface between PDEGMA and PDIPAEMA blocks. In general, lower molecular mass increases translational entropy of the chains, thus contributing to miscibility and suppressing segregation.  

Different End Groups and Functional Groups: Currently, the polymers have carboxyl end groups. It would be interesting to explore how amine end groups, or other functional groups with different pKa values and polarities, might influence the polymer's behavior. The authors should consider this aspect or suggest a systematic study to investigate the effects of different terminal groups.

We thank the reviewer for this suggestion. It would certainly be interesting to compare behavior of copolymers with different terminal end groups, however, it would probably require a different synthetic method than RAFT. In RAFT, the choice of terminal groups is dictated by the optimal CTA agent for the given synthesis. However, since RAFT is a very popular polymerization technique, it is useful to point out that end groups can affect the solution behavior, especially in the case of shorter polymers. 

Reviewer 3 Report

Comments and Suggestions for Authors

The work represents a continuation of research on block copolymers, which have already been the subject of several works.

The article needs to increase the evidence base in order to be published in the journal "Polymers". I have highlighted some comments and suggestions below that might help the authors:

1.      The expression "RAFT chain transfer agent" is incorrect (reversible addition-fragmentation chain transfer chain transfer agent).

2.      The introduction needs to be strengthened.

3.      The scientific significance and practical value of this work should be indicated.

4.      It is necessary to add information about the pH-meter used and conditions for determining pH.

5.      In Figure 1, the point of the second phase transition (318 K) should be indicated.

6.      Link missing [27]

7.      Since PDEGMA-PDIPAEMA is the diblock copolymer, it is better to use PDEGMA-b-PDIPAEMA

8.      Mw, PDI and composition for the block copolymer should be confirmed.

9.      The term "molecular weights" is usually used to indicate the molecular weight of polymers.

10.  Materials must be added to the "Materials and Methods"

11.  The expression "by the dehydration of the PDEGMA block" needs clarification (146).

12.  The expression "most likely caused by the formation of hydrogen bonds between COOH and CH2CH2O groups" requires confirmation (164). Interaction among these groups requires serious evidence.

13.  Scheme 2 is not informative.

14.  The authors do not discuss the behavior of PDIPAEMA with changes in acidity.

15.  The authors do not discuss the possibility of intermacromolecular association due to the electrostatic interaction of ionized carboxyl groups of PDEGMA and protonated PDIPAEMA.

Author Response

The work represents a continuation of research on block copolymers, which have already been the subject of several works.

The article needs to increase the evidence base in order to be published in the journal "Polymers". I have highlighted some comments and suggestions below that might help the authors:

  1.     The expression "RAFT chain transfer agent" is incorrect (reversible addition-fragmentation chain transfer chain transfer agent).

Thank you for pointing this out. We rephrased this expression.

  1.     The introduction needs to be strengthened.

We extended the introduction by a paragraph about properties of schizophrenic block copolymers and about schizophrenic micellization.

  1.     The scientific significance and practical value of this work should be indicated.

We extended the Conclusion with a comment pointing out that the proven ability of PDEGMA-b-PDIPAEMA associates to solubilize hydrophobic substances is of practical importance, with the advantage of spontaneously formation which does not require use of cosolvents as is the case of polymer nanoparticles prepared by nanoprecipitation. 

  1.     It is necessary to add information about the pH-meter used and conditions for determining pH.

We added this information to the experimental section.

  1.     In Figure 1, the point of the second phase transition (318 K) should be indicated.

We did it in the revised version of the manuscript (after a more precise reading, it is in fact 319 K, 318 K was only an estimate).

  1.     Link missing [27]

Thank you for the correction, the missing reference was added to the list.

  1.     Since PDEGMA-PDIPAEMA is the diblock copolymer, it is better to use PDEGMA-b-PDIPAEMA

As suggested, we changed the acronym PDEGMA-PDIPAEMA to PDEGMA-b-PDIPAEMA

  1.     Mw, PDI and composition for the block copolymer should be confirmed.

In the revised manuscript, we briefly overview information about characterization of the copolymer, including the SEC chromatogram and 1H and IR spectra.

  1.     The term "molecular weights" is usually used to indicate the molecular weight of polymers.

  1. Materials must be added to the "Materials and Methods"

As suggested, we added the subsection Materials into the Material and Methods section.

  1. The expression "by the dehydration of the PDEGMA block" needs clarification (146).

The phase separation of thermoresponsive polymers above LCST is caused by dehydration, that is, by a loss of water molecules solvating the polymer chain. We added this clarification to the revised manuscript.

  1. The expression "most likely caused by the formation of hydrogen bonds between COOH and CH2CH2O groups" requires confirmation (164). Interaction among these groups requires serious evidence.

Unfortunately, we do not have direct evidence for that interaction. Obtaining any, for instance from NMR spectroscopy, would be very difficult considering that there is only one COOH group per one chain. Our interpretation of PDEGMA-b-PDIPAEMA behavior is based on the existing firm evidence of the hydrogen bonding between polyacrylic (or polymethacrylic acid) with poly(ethylene oxide) in the literature.

  1. Scheme 2 is not informative.

We attempted to improve the Scheme 2 by adding also dodecyl terminal groups and showing also their influence on the association behavior and 

  1. The authors do not discuss the behavior of PDIPAEMA with changes in acidity.

In the revised manuscript, we enhance the discussion regarding the influence of the degree of protonation of PDIPAEMA on the association behavior. Moreover, we add new experimental data from zeta potential measurement at pH 2 and pH 7. 

  1. The authors do not discuss the possibility of intermacromolecular association due to the electrostatic interaction of ionized carboxyl groups of PDEGMA and protonated PDIPAEMA.

In the revised manuscript, we discuss this issue in detail. Briefly, zeta potential measurements showed that the surface of the associates was negatively charged, indiating that COO groups were exposed on that surface instead of being in the complex with protonated amino groups of PDIPAEMA. One COO– group as a monovalent ion is clearly not sufficient for proper electrostatic complexation with PDIPAEMA as it replaces only one monovalent counterion of PDIPAEMA, providing no entropic gain.

Round 2

Reviewer 2 Report

Comments and Suggestions for Authors

The authors have made substantial improvements to the manuscript, addressing all major concerns raised during the initial review. The revised manuscript is well-written, scientifically sound, and provides significant insights into the association behavior of thermoresponsive diblock copolymers in aqueous solutions. The enhanced experimental details, comprehensive data analysis, and thoughtful discussions make the study robust and valuable to the field.

I recommend the manuscript for publication in its current form.

Comments on the Quality of English Language

The quality of English in the revised manuscript is very good, with clear and concise language. Only minor revisions, such as checking for typographical errors and ensuring consistent tense usage, are needed to further improve readability.

Reviewer 3 Report

Comments and Suggestions for Authors

The authors did a great job in the revision process. The paper could be published in the current form.